# New Amino Acid Schiff Bases as Anticancer Agents via Potential Mitochondrial Complex I-Associated Hexokinase Inhibition and Targeting AMP-Protein Kinases/mTOR Signaling Pathway

**DOI:** 10.3390/molecules26175332

**Published:** 2021-09-02

**Authors:** Ahmed A. Noser, Aboubakr H. Abdelmonsef, Mohamed El-Naggar, Maha M. Salem

**Affiliations:** 1Organic Chemistry Division, Chemistry Department, Faculty of Science, Tanta University, Tanta 31527, Egypt; ahmed.nosir@science.tanta.edu.eg; 2Chemistry Department, Faculty of Science, South Valley University, Qena 83523, Egypt; 3Chemistry Department, Faculty of Sciences, University of Sharjah, Sharjah 27272, United Arab Emirates; melnagrr@sharjah.ac.ae; 4Biochemistry Division, Chemistry Department, Faculty of Science, Tanta University, Tanta 31527, Egypt; maha_salem@science.tanta.edu.eg

**Keywords:** cancer, hexokinase, NADH: ubiquinone oxidoreductase, docking study, cytotoxic activity

## Abstract

Two series of novel amino acid Schiff base ligands containing heterocyclic moieties, such as quinazolinone **3**–**11** and indole **12**–**20** were successfully synthesized and confirmed by spectroscopic techniques and elemental analysis. Furthermore, all compounds were investigated in silico for their ability to inhibit mitochondrial NADH: ubiquinone oxidoreductase (complex I) by targeting the AMPK/mTOR signaling pathway and inhibiting hexokinase, a key glycolytic enzyme to prevent the Warburg effect in cancer cells. This inhibitory pathway may be an effective strategy to cause cancer cell death due to an insufficient amount of ATP. Our results revealed that, out of 18 compounds, two (**11** and **20**) were top-ranked as they exhibited the highest binding energies of −8.8, −13.0, −7.9, and −10.0 kcal/mol in the docking analysis, so they were then selected for in vitro assessment. Compound **11** promoted the best cytotoxic effect on MCF-7 with IC_50_ = 64.05 ± 0.14 μg/mL (0.135 mM) while compound **20** exhibited the best cytotoxic effect on MDA-231 with IC_50_ = 46.29 ± 0.09 μg/mL (0.166 mM) Compounds **11** and **20** showed significant activation of AMPK protein and oxidative stress, which led to elevated expression of p53 and Bax, reduced Bcl-2 expression, and caused cell cycle arrest at the sub-G_0_/G_1_ phase. Moreover, compounds **11** and **20** showed significant inhibition of the mTOR protein, which led to the activation of aerobic glycolysis for survival. This alternative pathway was also blocked as compounds **11** and **20** showed significant inhibitory effects on the hexokinase enzyme. These findings demonstrate that compounds **11** and **20** obeyed Lipinski’s rule of five and could be used as privileged scaffolds for cancer therapy via their potential inhibition of mitochondrial complex I-associated hexokinase.

## 1. Introduction

Cancer represents a major global public health problem and is still associated with significant mortality [1]. Given the widespread occurrence of cancer drug resistance and a lack of sensitivity of tumor cells to such drugs, there is an urgent need for novel, effective, and less harmful antitumor agents that act by inhibiting specific metabolic target proteins [2]. Particularly important in this context is the synthesis of bioenergetic drugs that not only impact ATP production but also disrupt the biosynthetic pathway that relies on precursor metabolites found in the generation of ATP as required for cancer cell proliferation [3]. NADH: ubiquinone oxidoreductase (complex I) is the entry point of reduced NADH into the electron transport chain. Direct inhibition of complex I in cancer cells decreases the proton gradient and mitochondrial oxygen consumption rate [4], diminishes tricarboxylic acid cycle (TCA) activity and metabolites [4,5], and leads to decreased cellular ATP levels [6,7,8].

The inhibition of mitochondrial respiration and ATP production results in a compensatory increase in glycolysis through the Warburg effect (aerobic glycolysis) to restore cellular ATP levels required for the survival of cancer cells. The inhibition of hexokinase (ATP: D-hexose 6-phosphotransferase), a key enzyme that catalyzes the first step in the glycolysis pathway, prevents the rerouting of metabolic flux and leads to the depletion of cellular ATP. When ATP cannot meet the requirements of cancer cells and oxidative stress occurs, which promotes reactive oxygen species (ROS) generation, AMP-protein kinases (AMPK) become activated, and the mitochondrial membrane potential becomes dysfunctional [8,9,10]. Phosphorylation and activation of AMPK leads to the activation of p53 and Bax, arrests cell cycle progression, inactivates mammalian target of rapamycin (mTOR), decreases protein synthesis, and decreases the transcription of gluconeogenic genes.

Many efforts have been made to synthesize novel effective and selective anticancer drugs for complex I and hexokinase inhibition as they can worsen the cellular energy status, leading to a global decrease in ATP-consuming processes, and also induce oxidative stress. In proliferating cells, this can elicit a cytostatic state that is associated with reduced proliferation, explaining some clinical observations of decreased progression of cancer cell growth. Cancer cells that cannot eventually compensate for this reduced energy status and are affected by ROS generation may undergo apoptosis [11,12].

The Schiff base compounds (imines) containing heterocyclic systems, such as quinazolinone and indole nuclei, azomethine linkages, and phenyl rings, have attracted particular attention due to their potential applications in medicinal and pharmaceutical chemistry [13,14].

Quinazolinone and indole are good pharmacophoric scaffolds found in many biologically active compounds ranging from natural products to synthetic pharmaceutical drugs. These compounds are known to be associated with antimalarial, antimicrobial, anticonvulsant, and anticancer effects [15,16,17,18,19].

Given the importance of imines (Schiff bases) in the medicinal field and the continuous efforts by our group to develop novel anticancer scaffolds [2,20,21,22,23,24,25,26,27,28], the aim of this study was first to design and synthesize novel quinazolinone and indole amino acid Schiff bases. The synthesized compounds were then subjected to computer-based docking studies to investigate their binding modes of interaction with the active site of the target enzymes. In addition, they underwent in silico physicochemical and pharmacokinetic investigations to predict their absorption, distribution, metabolism, excretion, and toxicity (ADMET) properties, as well as analysis of the structure–activity relationship (SAR) of the compounds. Finally, the compounds were examined in vitro for inhibitory activities against mitochondrial NADH: ubiquinone oxidoreductase (complex I) by targeting the AMPK/mTOR signaling pathway and inhibiting hexokinase, a key glycolytic enzyme, to prevent the Warburg effect in cancer cells.

## 2. Materials and Methods

### 2.1. Chemicals and Drugs

Anthranilic acid, benzoyl chloride, p-aminoacetophenone, indol-3-carboxaldehyde, pyridine, triethylamine, glycine, phenylglycine, alanine, phenylalanine, serine, tyrosine, leucine, lysine, asparagine, thiobarbituric acid, trichloroacetic acid (TCA), reduced glutathione (GSH), and sodium pyrophosphate were purchased from Sigma-Aldrich Chemical Co. (St. Louis, MO, USA). Tamoxifen was purchased from EIMC United Pharmaceutics (Cairo, Egypt).

#### 2.1.1. General Information

Reactions were monitored by TLC performed on pre-coated plates Merck Kieselgel 60 F254 (EMD Millipore Corporation, Billerica, MA, USA). Infrared spectra were recorded at the central laboratory of Tanta University using a PerkinElmer 1420 spectrophotometer (Waltham, MA, USA). The spectra were obtained using the KBr disc technique. The samples were dried in an oven and then mounted on a sample holder with a large cavity. Melting points were determined by the open capillary method using the Gallenkamp melting point and are reported uncorrected. The elemental analyses of compounds were performed at the microanalytical center of Cairo University using a PerkinElmer 240 CHN Elemental analyzer (Waltham, MA, USA). ^1^H-NMR and ^13^C-NMR spectra were collected at a resonance frequency of 400 MHz at Kafr El-Sheikh University. NMR spectra were obtained on a Bruker AMC instrument (Bruker Biosciences Corporation, Billerica, MA, USA) operating at 400 MHz using dimethyl sulfoxide (DMSO) as a solvent and tetramethylsilane as an internal standard. The chemical shifts for ^1^H-NMR are reported in ppm from tetramethylsilane (0 ppm) or referenced to the solvent (DMSO-d_6_, δ 2.50). Chemical shifts (δ) for ^13^C-NMR spectra are referenced to the signals for residual deuterated solvents (DMSO-d_6_, 37.5). Multiplicities are reported using the following abbreviations: s (singlet), d (doublet), t (triplet), and m (multiplet).

#### 2.1.2. Synthesis of 2-Phenyl-4*H*-benzo[d][1,3] Oxazin-4-one (**1**)

Compound **1** was prepared as described by Tiwary [29] with an 86% yield.

#### 2.1.3. Synthesis of 3-(4-Acetylphenyl)-2-phenylquinazolin-4(3*H*)-one (**2**)

Compound **2** was prepared as described by Patel [30] with an 82% yield.

#### 2.1.4. Synthesis of Amino Acid Schiff Bases


**General method:**


The Schiff bases **3**–**20** were synthesized by the condensation of a carbonyl compound (3-(4-acetylphenyl)-2-phenylquinazolin-4(3*H*)-one and/or indol-3-carboxaldehyde) (1 mmol) and derivatives of amino acids (1 mmol) after stirring for 30 min with Et_3_N dissolved in dry ethanol (10 mL). The resulting reaction mixture was stirred and refluxed for 9 h (TLC control) and then allowed to cool overnight. The precipitated Schiff bases were filtered, washed with cold ethanol several times, and dried at room temperature. The solid products were then recrystallized with ethanol [31].

##### 2-(1-(4-(4-oxo-2-Phenylquinazolin-3(4*H*)-yl) phenyl) ethylideneamino) Acetic Acid (**3**)

Yield 82%; mp 146 °C; ^1^H NMR (400 MHz, DMSO-d_6_) δ (ppm): 11.00 (s, 1H, COOH), 7.08–7.95 (m, 13H, Ar-H), 4.49 (s, 2H, CH_2_), 0.96 (s, 3H, CH_3_); ^13^C NMR (400 MHz, DMSO-d_6_) δ (ppm): 171.60, 164.60, 164.20, 161.10, 151.50, 135.90, 135.30, 133.70, 130.40, 130.00, 129.60, 129.10, 128.90, 127.60, 126.30, 122.60, 121.90, 121.10, 54.70, 22.30; IR (KBr) ν: 3027 (arom. CH strech), 2953 (aliph. CH strech), 2900 (OH), 1715 (CO), 1610–1655 (C=N); Anal. Calcd for C_24_H_19_N_3_O_3_ (397.43): C, 72.53%; H, 4.82%; O, 12.08%; N, 10.57%. Found: C, 72.26%; H, 4.61%; N, 10.36%.

##### 2-(1-(4-(4-oxo-2-Phenylquinazolin-3(4*H*)-yl) phenyl) ethylideneamino) Propionic Acid (**4**)

Yield 80%; mp 191 °C; ^1^H-NMR (400 MHz, DMSO-d_6_) δ (ppm): 11.00 (s, 1H, COOH), 7.08–7.95 (m, 13H, Ar-H), 4.18 (m, 1H, CH), 1.42 (d, 3H, CH_3_), 0.96 (s, 3H, CH_3_); ^13^C-NMR (400 MHz, DMSO-d_6_) δ (ppm): 177.70, 164.60, 164.20, 161.10, 151.50, 135.90, 135.30, 133.70, 130.40, 130.00, 129.60, 129.10, 128.90, 127.60, 126.30, 122.60, 121.90, 121.10, 62.80, 22.60, 18.20; IR (KBr) ν: 3027 (arom. CH strech), 2956 (aliph. CH strech), 2915 (OH), 1725 (CO), 1615–1659 (C=N); Anal. Calcd for C_25_H_21_N_3_O_3_ (411.45): C, 72.98%; H, 5.14%; O, 11.67%; N, 10.21%. Found: C, 72.36%; H, 4.91%; N, 10.32%.

##### 2-(1-(4-(4-oxo-2-Phenylquinazolin-3(4*H*)-yl) phenyl) ethylidene amino)-3-phenyl Propionic Acid (**5**)

Yield 84%; mp 165 °C; ^1^H-NMR (400 MHz, DMSO-d_6_) δ (ppm): 11.00 (s, 1H, COOH), 6.81–7.95 (m, 18H, Ar-H), 4.39 (t, 1H, CH), 3.29 (d, 2H, CH_2_), 0.96 (s, 3H, CH_3_); ^13^C-NMR (400 MHz, DMSO-d_6_) δ (ppm): 177.70, 164.60, 164.20, 161.10, 151.50, 139.70, 135.90, 135.30, 133.70, 130.40, 130.00, 129.60, 129.10, 128.90, 128.00, 127.60, 126.30, 126.20, 122.60, 121.90, 121.10, 66.80, 38.40, 22.60; IR (KBr) ν: 3037 (arom. CH strech), 2956 (aliph. CH strech), 1715 (CO), 2912 (OH), 1612–1657 (C=N); Anal. Calcd for C_31_H_25_N_3_O_3_ (487.55): C, 76.37%; H, 5.17%; O, 9.84%; N, 8.62%. Found: C, 76.11%; H, 4.99%; N, 8.46%.

##### 3-Hydroxy-2-(1-(4-(4-oxo-2-phenylquinazolin-3(4*H*)-yl) phenyl) ethylidene amino) Propionic Acid (**6**)

Yield 80%; mp 184 °C; ^1^H-NMR (400 MHz, DMSO-d_6_) δ (ppm): 11.00 (s, 1H, COOH), 7.08–7.95 (m, 13H, Ar-H), 4.25 (d, 2H, CH_2_), 4.09 (t, 1H, CH), 2.00 (s, 1H, OH), 0.96 (s, 3H, CH_3_); ^13^C-NMR (400 MHz, DMSO-d_6_) δ (ppm): 177.70, 164.60, 164.20, 161.10, 151.50, 135.90, 135.30, 133.70, 130.40, 130.00, 129.60, 129.10, 128.90, 127.60, 126.30, 122.60, 121.90, 121.10, 66.20, 64.00, 22.60; IR (KBr) ν: 3250 (CH_2_-OH), 3027 (arom. CH strech), 2953 (aliph. CH strech), 2900 (carboxylic-OH), 1715 (CO), 1610–1655 (C=N); Anal. Calcd for C_25_H_21_N_3_O_4_ (427.45): C, 70.25%; H, 4.95%; O, 14.97%; N, 9.83%. Found: C, 69.46%; H, 4.76%; N, 9.66%.

##### 3-(4-Hydroxyphenyl)-2-(1-(4-(4-oxo-2-phenylquinazolin-3(4*H*)-yl) phenyl) ethylidene amino) Propionic Acid (**7**)

Yield 86%; mp 150 °C; ^1^H NMR (400 MHz, DMSO-d_6_) δ (ppm): 11.00 (s, 1H, COOH), 7.08–7.95 (m, 13H, Ar-H), 6.61–6.75 (m, 4H, Ar-H), 5.00 (s, 1H, OH), 4.39 (t, 1H, CH), 3.22 (d, 2H, CH_2_), 0.96 (s, 3H, CH_3_); ^13^C NMR (400 MHz, DMSO-d_6_) δ (ppm): 178.10, 164.80, 164.20, 161.10, 157.50, 151.50, 135.90, 135.30, 133.70, 131.30, 130.40, 130.00, 129.60, 129.10, 128.90, 127.60, 126.30, 122.60, 121.90, 121.10, 116.50, 66.80, 38.60, 16.60; IR (KBr) ν: 3290 (arom-OH), 3027 (arom. CH strech), 2953 (aliph. CH strech), 2900 (carboxylic-OH), 1715 (CO), 1610–1655 (C=N); Anal. Calcd for C_31_H_25_N_3_O_4_ (503.55): C, 73.94%; H, 5.00%; O, 12.71%; N, 8.34%. Found: C, 73.16%; H, 4.83%; N, 8.16%.

##### 4-Methyl-2-(1-(4-(4-oxo-2-phenylquinazolin-3(4*H*)-yl) phenyl) ethylideneamino) Pentanoic Acid (**8**)

Yield 84%; mp 154 °C; ^1^H-NMR (400 MHz, DMSO-d_6_) δ (ppm): 11.00 (s, 1H, COOH), 7.08–7.95 (m, 13H, Ar-H), 4.00 (t, 1H, CH), 1.85 (t, 2H, CH_2_), 1.83 (m, 1H, CH), 1.15 (d, 6H, 2CH_3_), 0.96 (s, 3H, CH_3_); ^13^C-NMR (400 MHz, DMSO-d_6_) δ (ppm): 177.70, 164.80, 164.20, 161.10, 151.50, 135.90, 135.30, 133.70, 130.40, 130.00, 129.60, 129.10, 128.90, 127.60, 126.30, 122.60, 121.90, 121.10, 56.90, 23.50, 23.20, 23.10, 16.60; IR (KBr) ν: 3027 (arom. CH strech), 2953 (aliph. CH strech), 2900 (OH), 1715 (CO), 1610–1655 (C=N); Anal. Calcd for C_28_H_27_N_3_O_3_ (453.53): C, 74.15%; H, 6.00%; O, 10.58%; N, 9.27%. Found: C, 73.26%; H, 5.81%; N, 8.36%.

##### 6-Amino-2-(1-(4-(4-oxo-2-phenylquinazolin-3(4*H*)-yl) phenyl) ethylideneamino) Hexanoic Acid (**9**)

Yield 80%; mp 193 °C; ^1^H-NMR (400 MHz, DMSO-d_6_) δ (ppm): 11.00 (s, 1H, COOH), 7.08–7.95 (m, 13H, Ar-H), 4.00 (t, 1H, CH), 2.71 (t, 2H, N-CH_2_), 2.00 (s, 2H, NH_2_), 1.88 (m, 2H, CH-CH_2_), 1.29–1.38 (m, 4H, 2CH_2_), 0.96 (s, 3H, CH_3_); ^13^C-NMR (400 MHz, DMSO-d_6_) δ (ppm):177.70, 164.80, 164.20, 161.10, 151.50, 135.90, 135.30, 133.70, 130.40, 130.00, 129.60, 129.10, 128.90, 127.60, 126.30, 122.60, 121.90, 121.10, 59.70, 42.30, 38.30, 32.70, 16.60, 12.80; IR (KBr) ν: 3390 (NH_2_), 3027 (arom. CH strech), 2953 (aliph. CH strech), 2900 (OH), 1715 (CO), 1610–1655 (C=N); Anal. Calcd for C_28_H_28_N_4_O_3_ (468.55): C, 71.78%; H, 6.02%; O, 10.24%; N, 11.96%. Found: C, 71.16%; H, 5.81%; N, 11.46%.

##### 4-Amino-4-oxo-2-(1-(4-(4-oxo-2-phenylquinazolin-3(4*H*)-yl) phenyl) ethylidene amino) Butanoic Acid (**10**)

Yield 87%; mp 140 °C; ^1^H-NMR (400 MHz, DMSO-d_6_) δ (ppm): 11.00 (s, 1H, COOH), 7.08–7.95 (m, 13H, Ar-H), 6.00 (s, 2H, NH_2_), 4.23 (t, 1H, CH), 2.64–2.77 (d, 2H, CH_2_), 0.96 (s, 3H, CH_3_); ^13^C-NMR (400 MHz, DMSO-d_6_) δ (ppm): 177.70, 174.80, 164.80, 164.20, 161.10, 151.50, 135.90, 135.30, 133.70, 130.40, 130.00, 129.60, 129.10, 128.90, 127.60, 126.30, 122.60, 121.90, 121.10, 56.50, 37.50, 16.60; IR (KBr) ν: 3390 (NH_2_), 3027 (arom. CH strech), 2953 (aliph. CH strech), 2900 (OH), 1715 (CO), 1610–1655 (C=N); Anal. Calcd for C_26_H_22_N_4_O_4_ (454.48): C, 68.71%; H, 4.88%; O, 14.08%; N, 12.33%. Found: C, 68.14%; H, 4.71%; N, 11.96%.

##### 2-(1-(4-(4-oxo-2-Phenylquinazolin-3(4*H*)-yl) phenyl) ethylideneamino)-2-phenyl Acetic Acid (**11**)

Yield 87%; mp 196 °C; ^1^H NMR (400 MHz, DMSO-d_6_) δ (ppm): 11.00 (s, 1H, COOH), 7.08–7.95 (m, 13H, Ar-H), 6.84–6.95 (m, 5H, Ar-H), 5.32 (s, 1H, CH), 0.96 (s, 3H, CH_3_); ^13^C NMR (400 MHz, DMSO-d_6_) δ (ppm): 178.20, 164.80, 164.20, 161.10, 151.50, 138.70, 135.90, 135.30, 133.70, 130.40, 130.00, 129.60, 129.40, 129.10, 128.90, 127.80, 127.60, 126.30, 122.60, 121.90, 121.10, 64.70, 22.60; IR (KBr) ν: 3027 (arom. CH strech), 2953 (aliph. CH strech), 2900 (OH), 1715 (CO), 1610–1655 (C=N); Anal. Calcd for C_30_H_23_N_3_O_3_ (473.52): C, 76.09%; H, 4.90%; O, 10.14%; N, 8.87%. Found: C, 75.26%; H, 4.74%; N, 8.16%.

##### 2-((1*H*-Indol-3-yl) methyleneamino) Acetic Acid (**12**)

Yield 92%; mp 175 °C; ^1^H-NMR (400 MHz, DMSO-d_6_) δ (ppm): 11.00 (s, 1H, COOH), 9.50 (s, 1H, NH), 7.50 (s, 1H, CH=N), 7.11–7.58 (m, 5H, Ar-H), 2.31 (s, 2H, CH_2_); ^13^C-NMR (400 MHz, DMSO-d_6_) δ (ppm): 171.60, 161.10, 135.70, 131.00, 126.30, 122.40, 120.30, 119.20, 111.30, 102.20, 55.20; IR (KBr) ν: 3027 (arom. CH strech), 2953 (aliph. CH strech), 2915 (OH), 1715 (CO), 1610–1655 (C=N); Anal. Calcd for C_11_H_10_N_2_O_2_ (202.21): C, 65.34%; H, 4.98%; O, 15.82%; N, 13.85%. Found: C, 65.11%; H, 4.65%; N, 13.24%.

##### 2-((1*H*-Indol-3-yl)methyleneamino) Propionic Acid (**13**)

Yield 87%; mp 120 °C; ^1^H-NMR (400 MHz, DMSO-d_6_) δ (ppm): 11.00 (s, 1H, COOH), 9.50 (s, 1H, NH), 7.50 (s, 1H, CH=N), 7.11–7.58 (m, 5H, Ar-H), 2.65 (m, 1H, CH), 1.11 (d, 3H, CH_3_); ^13^C-NMR (400 MHz, DMSO-d_6_) δ (ppm):171.60, 161.10, 135.70, 131.00, 126.30, 122.40, 120.30, 119.20, 111.30, 102.20, 55.20; IR (KBr) ν: 3037 (arom. CH strech), 2966 (aliph. CH strech), 2925 (OH), 1715 (CO), 1615–1649 (C=N); Anal. Calcd for C_12_H_12_N_2_O_2_ (216.24): C, 66.65%; H, 5.59%; O, 14.80%; N, 12.96%. Found: C, 66.39%; H, 5.24%; N, 12.64%.

##### 2-((1*H*-Indol-3-yl) methyleneamino)-3-phenylpropionic Acid (**14**)

Yield 89%; mp 180 °C; ^1^H-NMR (400 MHz, DMSO-d_6_) δ (ppm): 11.00 (s, 1H, COOH), 9.50 (s, 1H, NH), 7.50 (s, 1H, CH=N), 7.11–7.81 (m, 10H, Ar-H), 3.11 (d, 2H, CH_2_), 2.89 (t, 1H, CH); ^13^C-NMR (400 MHz, DMSO-d_6_) δ (ppm): 177.70, 161.10, 139.70, 135.70, 131.00, 128.90, 128.00, 126.30, 126.20, 122.40, 120.30, 119.20, 111.30, 102.20, 67.30, 38.30; IR (KBr) ν: 3037 (arom. CH strech), 2956 (aliph. CH strech), 2923 (OH), 1716 (CO), 1615–1655 (C=N); Anal. Calcd for C_18_H_16_N_2_O_2_ (292.33): C, 73.95%; H, 5.52%; O, 10.95%; N, 9.58%. Found: C, 73.21%; H, 5.27%; N, 9.17%.

##### 2-((1*H*-Indol-3-yl) methyleneamino)-3-hydroxy Propionic Acid (**15**)

Yield 91%; mp 163 °C; ^1^H-NMR (400 MHz, DMSO-d_6_) δ (ppm): 11.00 (s, 1H, COOH), 9.50 (s, 1H, NH), 7.50 (s, 1H, CH=N), 7.11–7.58 (m, 5H, Ar-H), 3.90 (d, 2H, CH_2_), 2. 65 (t, 1H, CH), 2.00 (s, 1H, OH); ^13^C-NMR (400 MHz, DMSO-d_6_) δ (ppm): 177.70, 161.10, 135.70, 131.00, 126.30, 122.40, 120.30, 119.20, 111.30, 102.20, 66.70, 63.70; IR (KBr) ν: 3260 (CH_2_-OH), 3027 (arom. CH strech), 2953 (aliph. CH strech), 2900 (carboxylic-OH), 1715 (CO), 1610–1655 (C=N); Anal. Calcd for C_12_H_12_N_2_O_3_ (232.24): C, 62.06%; H, 5.21%; O, 20.67%; N, 12.06%. Found: C, 61.72%; H, 4.86%; N, 11.66%.

##### 2-((1*H*-Indol-3-yl) methyleneamino)-3-(4-hydroxyphenyl) Propionic Acid (**16**)

Yield 94%; mp 203 °C; ^1^H NMR (400 MHz, DMSO-d_6_) δ (ppm): ^1^H NMR (400 MHz, DMSO-d_6_) δ (ppm): 11.00 (s, 1H, COOH), 9.50 (s, 1H, NH), 7.50 (s, 1H, CH=N), 7.11–7.41 (m, 5H, Ar-H), 6.65–6.95 (m, 4H, Ar-H), 5.00 (s, 1H, OH), 3.71 (t, 1H, CH), 2.90 (d, 2H, CH_2_); ^13^C NMR (400 MHz, DMSO-d_6_) δ (ppm): 177.70, 161.10, 159.90, 135.70, 132.30, 131.00, 129.40, 126.30, 122.40, 120.30, 119.20, 116.00, 111.30, 102.20, 73.30, 38.30; IR (KBr) ν: 3290 (arom-OH), 3027 (arom. CH strech), 2953 (aliph. CH strech), 2900 (carboxylic-OH), 1715 (CO), 1610–1655 (C=N); Anal. Calcd for C_18_H_16_N_2_O_3_ (308.33): C, 70.12%; H, 5.23%; O, 15.57%; N, 9.09%. Found: C, 69.76%; H, 4.99%; N, 8.76%.

##### 2-((1*H*-Indol-3-yl) methyleneamino)-4-methylpentanoic Acid (**17**)

Yield 94%; mp 207 °C; ^1^H-NMR (400 MHz, DMSO-d_6_) δ (ppm): 11.00 (s, 1H, COOH), 9.50 (s, 1H, NH), 7.50 (s, 1H, CH=N), 7.11–7.55 (m, 5H, Ar-H), 2.40 (t, 1H, CH), 1.83 (m, 1H, CH), 1.55 (t, 2H, CH_2_), 1.00 (d, 6H, 2CH_3_); ^13^C-NMR (400 MHz, DMSO-d_6_) δ (ppm): 177.70, 170.10, 135.70, 131.00, 126.30, 122.40, 120.30, 119.20, 111.30, 102.20, 63.40, 23.20, 23.10; IR (KBr) ν: 3027 (arom. CH strech), 2953 (aliph. CH strech), 2900 (OH), 1715 (CO), 1610–1655 (C=N); Anal. Calcd for C_15_H_18_N_2_O_2_ (258.32): C, 69.74%; H, 7.02%; O, 12.39%; N, 10.84%. Found: C, 69.38%; H, 6.76%; N, 10.46%.

##### 2-((1*H*-Indol-3-yl) methyleneamino)-6-aminohexanoic Acid (**18**)

Yield 86%; mp 114 °C; ^1^H-NMR (400 MHz, DMSO-d_6_) δ (ppm): 11.00 (s, 1H, COOH), 9.50 (s, 1H, NH), 7.50 (s, 1H, CH=N), 7.11–7.55 (m, 5H, Ar-H), 2.60 (t, 2H, N-CH_2_), 2.30 (t, 1H, CH), 2.00 (s, 2H, NH_2_), 1.70 (m, 2H, CH-CH_2_), 1.51 (m, 4H, 2CH_2_); ^13^C-NMR (400 MHz, DMSO-d_6_) δ (ppm): 177.70, 161.10, 135.70, 131.00, 126.30, 122.40, 120.30, 119.20, 111.30, 102.20, 66.20, 61.80, 42.30, 38.00, 32.70; IR (KBr) ν: 3380 (NH_2_), 3027 (arom. CH strech), 2953 (aliph. CH strech), 2900 (OH), 1715 (CO), 1610–1655 (C=N); Anal. Calcd for C_15_H_19_N_3_O_2_ (273.33): C, 65.91%; H, 7.01%; O, 11.71%; N, 15.37%. Found: C, 65.45%; H, 6.75%; N, 15.16%.

##### 2-((1*H*-Indol-3-yl)methyleneamino)-4-amino-4-oxobutanoic Acid (**19**)

Yield 93%; mp 181 °C; ^1^H-NMR (400 MHz, DMSO-d_6_) δ (ppm): 11.00 (s, 1H, COOH), 9.50 (s, 1H, NH), 7.50 (s, 1H, CH=N), 7.11–7.55 (m, 5H, Ar-H), 6.00 (s, 2H, NH_2_), 2.65 (t, 1H, CH), 2.31–2.41 (d, 2H, CH_2_); ^13^C-NMR (400 MHz, DMSO-d_6_) δ (ppm): 177.70, 174.80, 161.10, 135.70, 131.00, 126.30, 122.40, 120.30, 119.20, 111.30, 102.20, 63.00, 37.20; IR (KBr) ν: 3390 (NH_2_), 3027 (arom. CH strech), 2953 (aliph. CH strech), 2900 (OH), 1715 (CO), 1610–1655 (C=N); Anal. Calcd for C_13_H_13_N_3_O_3_ (259.26): C, 60.22%; H, 5.05%; O, 18.51%; N, 16.21%. Found: C, 59.88%; H, 4.82%; N, 15.96%.

##### 2-((1*H*-indol-3-yl)methyleneamino)-2-phenylacetic Acid (**20**)

Yield 94%; mp 166 °C; ^1^H NMR (400 MHz, DMSO-d_6_) δ (ppm): ^1^H NMR (400 MHz, DMSO-d_6_) δ (ppm): 11.00 (s, 1H, COOH), 9.50 (s, 1H, NH), 7.50 (s, 1H, CH=N), 7.31–7.72 (m, 5H, Ar-H), 7.00–7.29 (m, 5H, Ar-H), 3.75 (s, 1H, CH); ^13^C NMR (400 MHz, DMSO-d_6_) δ (ppm): 178.20, 161.10, 138.70, 135.70, 131.00, 130.00, 129.40, 127.80, 126.30, 122.40, 120.30, 119.20, 111.30, 102.20, 65.20; IR (KBr) ν: 3027 (arom. CH strech), 2953 (aliph. CH strech), 2900 (OH), 1715 (CO), 1610–1655 (C=N); Anal. Calcd for C_17_H_14_N_2_O_2_ (278.31): C, 73.37%; H, 5.07%; O, 11.50%; N, 10.07%. Found: C, 72.95%; H, 4.87%; N, 9.46%.

### 2.2. In Silico Study

The three-dimensional coordinate files of the target enzymes hexokinase (1bdg) and NADH oxidoreductase (3m9s) were obtained from the Protein Data Bank (http://www.rscb.org/pdb/ accessed on 13 April 2021) [32]. The chemical structures of the new Schiff base compounds were sketched using ChemDraw Ultra 7.0 and then converted into SDF format using the Open Babel GUI tool [33]. Two in-house libraries of all the synthesized Schiff base molecules were generated for the docking process. The enzyme–ligand interaction study was performed using the PyRx virtual screening tool [34]. The Discovery Studio 3.5 tool was used to visualize the intermolecular interactions between the ligand molecules and enzymes. The prediction of molecular properties and drug-likeness of all synthesized compounds was achieved using the freely available tools Mol Inspiration, SwissADME, and admetSAR. Lipinski’s role of five (Ro5) was used to evaluate the drug-likeness of the prepared molecules [MW ≤ 500, HBA (2.0–20.0); HBD (0.0–6.0); logp ˂ 5; N rotatable ≤ 10; topological polar surface area (TPSA) ≤ 140; % (HIA^+^) > 80% high, <25% low; volume (500–2000)].

### 2.3. In Vitro Anticancer Studies on Predicted Compounds

The potent compounds in docking studies were selected to study their anticancer effect using MTT assay and then subjected to further analyses.

#### 2.3.1. Cell Culture, Maintenance, and Treatment

The estrogen receptor-positive breast cancer cell line MCF-7, triple-negative breast cancer cell line MDA-231, and pancreatic cancer cell line (PCL), along with the human normal epithelial amnion cell line WISH as a model for normal cells, were maintained and cultured in Dulbecco’s Modified Eagle’s Medium with 10% fetal bovine serum (all obtained from Gibco-BRL, New York, NY, USA) and 1% penicillin/streptomycin under a 5% CO_2_ and 95% humidified atmosphere at 37 °C in a CO_2_ incubator. All cells were provided by the National Cancer Institute (Cairo University, Giza Egypt). The cells were incubated for 48 h with selected compounds at different concentrations (0–200 g/mL) and tamoxifen (TAM) as a reference drug (0–100 g/mL) and then subjected to analysis.

#### 2.3.2. Cell Cytotoxicity Assay by MTT

In vitro cell viability was tested using the tetrazolium 3-(4,5-dimethylthiazol-2-yl)-2,5-diphenyl-tetrazolium bromide (MTT) assay. The cells were separately seeded in a 96-well plate (1 × 10^4^ cells/well, 100 μL/well) containing appropriate medium and incubated with the drug at different doses for 48 h. Then, the cells were incubated with 5 mg/mL MTT (Gibco-BRL, New York, NY, USA) for 4 h, followed by replacement of the medium with 100 μL of DMSO (Sigma-Aldrich, St. Louis, MO, USA) and vortexing for 20 min. Absorbance was recorded at 570 nm using a Model 680 microplate reader (Bio-Rad, Hercules, CA, USA) [35]. The concentration of the selected compound and TAM inhibiting 50% of cells (IC_50_) was calculated using the sigmoidal curve with GraphPad (Prism) statistical software.

#### 2.3.3. Cell Cycle Analysis

Cell cycle phase analysis was performed by flow cytometry following the method of Darzynkiewicz et al. [36] with some modifications. Briefly, following trypsinization, MCF-7 and MDA-231 cells were centrifuged at 4500 rpm for 5 min, washed twice, resuspended in warm PBS, fixed with ice-cold absolute ethanol, and then incubated at −20 °C for 24 h. After two washes with PBS, the cells were resuspended in propidium iodide (PI) solution containing 100 μL (0.05 mg/mL) of PI, 50 μL (0.2 mg/mL) of RNase A, and 0.1% *v*/*v* Triton X-100 in PBS and incubated in the dark for 30–60 min at room temperature. Finally, the pellet was washed with 1X PBS and resuspended in 300 µL 1X PBS and analyzed using an Accuri C6 flow cytometer (Becton Dickinson, Franklin Lake, BD, USA) with PE × FL2 channels.

#### 2.3.4. QPCR Analysis

The qPCR was carried out on treated and control cells to assess Bax, p53, and Bcl-2 mRNA expression. Briefly, total RNA was extracted using RNeasy Plus Minikit (Qiagen, Hilden, Germany) following the manufacturer’s protocol and as previously described by Kvastad et al. [37]. The quality of RNA was assessed by 1% agarose gel electrophoresis and the concentration was determined using Nanodrop (Q5000, Quawell, 1920 city, Chester, PA, USA). For cDNA synthesis, total RNA was reverse-transcribed with RevertAid H Minus Reverse Transcriptase following the manufacturer’s instructions (Thermo Fisher Scientific, Waltham, MA, USA). Real-time PCR was performed using Power SYBR Master Mix (Thermo Fisher Scientific) on an Applied Biosystems 7500 system (Applied Biosystems, Waltham, MA, USA), following the standard program of the reaction cycle of 95 °C for 10 min, followed by 40–45 cycles at 95 °C for 15 s and at 60 °C for 1 min, as recommended by the manufacturer. Samples of cDNA were run in triplicate. All data were then normalized to the endogenous control, GAPDH, a housekeeping gene. The quantity critical thresholds (Ct) of the target gene were normalized with the quantity (Ct) of GAPDH. Fold change in gene expression was calculated using the comparative threshold cycle (2-ΔΔCT) method of Livak and Schmittgen [38]. For the treated groups, 2-ΔΔCT assessment was used to determine the fold change in gene expression relative to the control untreated group. The primer sequences used in this study are provided in Table 1.

#### 2.3.5. Detection of Hexokinase Activity Level (ELISA)

Six-well tissue culture plates were seeded with 1 × 10^5^ MCF-7 and MDA-231 cells and left for 24 h under optimal culture conditions to obtain a confluent sheet of cells, which were then treated for 48 h. Subsequently, the cells were harvested, and their lysate was extracted to estimate the levels of hexokinase activity using human ELISA kits following the manufacturer’s instructions (cat. No. MAK091; Sigma-Aldrich).

#### 2.3.6. Estimation of Oxidative/Antioxidant Biomarkers

Six-well tissue culture plates were seeded with 1 × 10^5^ MCF-7 and MDA-231 cells and left for 24 h under optimal culture conditions to obtain a confluent layer of cells, which were then treated for 48 h. Subsequently, the cells were washed with PBS and scraped. The scraped cells were incubated in lysis buffer [20 mM Tris–HCl (pH 7.5), 150 mM NaCl, 1 mM Na_2_EDTA, 1% Triton, and 2.5 mM sodium pyrophosphate]. Then, the cells were centrifuged at 15,000 rpm and 4 °C for 15 min. The supernatant was used to measure the levels of malondialdehyde (MDA) and reduced glutathione (GSH) [39] and the protein concentration was measured using the Bradford assay [40].

#### 2.3.7. Immunoblotting Analysis

Immunoblotting was performed following the method reported by Mruk and Cheng [41]. In this method, the proteins are extracted from cells using ice-cold lysis buffer 50 mM Tris-HCl pH 7.4, 150 mM NaCl, 1% Nonidet P-40, 1 mM ethylenediaminetetraacetic acid (EDTA), 1X Protease Inhibitor Cocktail]. The protein was quantified using a Bradford assay kit (Thermo Scientific), and equal quantities of protein (20 µg) were resolved by 12% SDS-polyacrylamide gel electrophoresis and then transferred onto a polyvinylidene difluoride (PVDF) membrane. This membrane was blocked with 5% skimmed milk for 1 h at room temperature and then incubated with specific primary antibodies phospho-AMPK (2535) (1:1000) and phospho-mTOR (29771) (1:1000). Blots were incubated with horseradish peroxidase (HRP)-conjugated goat anti-rabbit IgG (H + L) (ab205718) (1:1000) and visualized using enhanced chemiluminescence (Pierce ECL Western blotting substrate) and Alliance gel doc (Alliance 4.7 Gel doc, Alliance, UK). UV Tec software (UK) was used to semi-quantify the protein bands. The density of each band was normalized using β-actin (SC-69879).

#### 2.3.8. Statistical Analysis

The experimental data are expressed as mean ± SE. The significance of differences among the various treated groups and control was analyzed using one-way ANOVA followed by Tukey’s test with GraphPad Prism 6 software (San Diego, CA, USA). Differences were considered to be significant at *p* < 0.05.

## 3. Results and Discussion

### 3.1. Chemistry of the Synthesized Compounds

2-Phenyl-4*H*-benzo[*d*][1,3]oxazin-4-one 1 was synthesized and characterized as previously described in the literature [29]. In the present study, the synthesis of 3-(4-acetylphenyl)-2-phenylquinazolin-4(3H)-one 2 was successfully achieved by refluxing compound 1 with *p*-amino acetophenone in the presence of anhydrous pyridine, to give a high yield of 82%, as shown in Scheme 1.

The suggested Schiff base compounds **3**–**11** were synthesized by a condensation reaction of compound **2** with various amino acids in the presence of triethylamine (TEA) and ethanol as solvent, as shown in Scheme 2.

Furthermore, Schiff bases **12**–**20** were synthesized by the condensation reaction of indol-3-carboxaldehyde with different amino acids in the presence of Et_3_N and ethanol as solvent, as shown in Scheme 3.

All synthesized compounds **3**–**20** were characterized by spectroscopic techniques, such as FT-IR and NMR spectroscopy, and elemental analysis as shown in the Appendix A. The FT-IR spectra of Schiff bases showed that the band of -C=N imine stretching vibration appeared for the Schiff bases in the range of 1610–1655 cm^−1^.

### 3.2. In Silico Docking Study

In this study, the binding mode of quinazolinone **3**–**11** and indole **12**–**20** Schiff base derivatives with the prospective crystal structures of hexokinase HK (PDB ID: 1bdg) and NADH oxidoreductase (PDB ID: 3m9s) enzymes was investigated. In silico molecular docking studies [42] were performed for all compounds using molecular docking in the PyRx tool, to explore and explain the orientation of molecules bound in the active sites of the target enzymes. The eighteen molecules showed good binding energy, indicating that they were successfully docked to the active site of the target enzymes HK and NADH [43]. Figure 1 and Figure 2 represent the molecular interactions between the best docked compounds (**11** and **20**), with the target enzymes. Different interactions, as well as the binding energies, are presented in Table 2. The 2D interactions of the other compounds are included in the Appendix A.


**Docking study against hexokinase enzyme**


For the Schiff base compounds containing quinazolinone moiety **3**–**11** (first library), the placement of compound **3** into a hexokinase active site exhibited HB and π-σ interactions with Glu220 and Pro105 at distances of 2.38 and 3.61 Å, respectively. Compounds **4**–**6** and **10** were stabilized in the pocket of the target enzyme through π-σ interactions with Met300 at 3.87, 3.88, 3.91, and 3.83 Å, respectively. In addition, compounds **4** and **6** formed HB with Ile114, and Tyr90 at 3.00 and 2.95 Å, respectively. Moreover, compound **11** with the highest binding energy of the first library (−8.8 kcal/mol) docked to the hexokinase enzyme through hydrogen bonds and π-cation interactions. It formed molecular interactions with the amino acid residues Leu349, Tyr373, Glu376, and Arg369 at distances of 2.91, 2.98, 2.16, 4.45, 3.78, 5.98, and 4.98 Å, respectively. 

On the other hand, regarding the Schiff base compounds containing indole moiety **12**–**20** (second library), all compounds except **13** and **19** formed HB interactions with residue Thr232. Compound 12 docked to the target through HB with residues Thr232 and Gly233 at 3.10, 2.93, and 2.83 Å, respectively. Compound 13 formed two HB with Gly303 and Thr336. Additionally, compounds **14**–**18** and **20** were completely enfolded in the active site forming similar interactions through the amino acid residues Thr88, Thr232, Gly233, Asp209, Asp413, Ser415, and Glu446. Furthermore, compounds **12**, **14**, **17**, and **18** exhibited π-cation interactions with the residue Lys418. Finally, molecule **20** with the highest binding energy of the second library (−7.9 kcal/mol) was successfully docked to the target enzyme hexokinase through HB interactions.


**Docking study against NADH oxidoreductase enzyme**


For the first library, all of compounds **3**–**11** docked to the target enzyme NADH oxidoreductase through four π-π interactions with residues Phe78 and Phe205. They also formed π-cation interactions with the residue Lys75, with the exceptions of **8** and **9**. In addition, they exhibited similar HB to Lys202. Compound **11** with the highest affinity to NADH oxidoreductase enzyme (−13.0 kcal/mol) formed one H-bond, four π-π, and two π-cation interactions with residues Lys202, Phe205, Phe78, and Lys75 at distances of 2.98, 4.33, 3.93, 4.40, 4.19, 5.21, and 5.33 Å, respectively. 

On the other hand, all of the compounds related to the Schiff bases containing indole moiety (the second library) **12**–**20** formed π-π interactions with Tyr180. In addition, they formed HB interactions with the target enzyme, though various residues, such as Asn92, Ser96, Ser100, Asp103, and Tyr180. Molecule **20** docked with NADH oxidoreductase enzyme with high affinity (−10.0 kcal/mol) through different types of non-covalent interaction, such as HB and π-π interactions with residue Tyr180 at 2.97 and 4.34 Å, respectively.


**ADMET properties and SAR analysis**


The ADMET data of two quinazolinone and indole Schiff base compounds were predicted as shown in Table 3. The results clearly pointed out that all compounds, except **12**–**14** and **17** cannot pass the blood–brain barrier (BBB), which confirmed their good CNS safety profile. Further, it was found that the compounds can be absorbed by the intestine. Their molecular weights were in the acceptable range (≤500 g/mol), with the exception of compound **7**. The results of in silico absorption percentage calculations showed high absorption percentage values (73–99%). Hence, it can be concluded that the compounds possess good absorption and distribution properties. 

The obtained TPSA values for molecules **3**–**20** were below 140 A^2^, indicating that the compounds had considerable permeability into the plasma membrane. All target compounds were in an acceptable range for HBD (1–4), HBA (4–8), and rotatable bonds (3–9). In addition, the compounds had high gastrointestinal (GI) absorption, confirming that they have excellent potential for absorption from the intestine after oral administration. All of the ligands had good bioavailability with scores ranging from 0.55 to 0.85, which is an indication that all compounds reached the circulatory system easily. Moreover, these compounds exhibited negative results on toxicity and carcinogenicity tests. The number of violations of Lipinski’s rule of five is listed in Table 3, which showed that all molecules fulfilled this rule with zero violations, with the exception of compound **7**, which represents the only Lipinski violation among these target compounds. Therefore, all molecules might be realistic drug candidates.

The structure-activity relationship (SAR) revealed why the combinations of Schiff base compounds (imines) with heterocyclic systems, such as quinazolinone and indole nucleus, azomethine linkage, and phenyl ring have attracted particular attention due to their potential applications in medicinal and pharmaceutical chemistry.

### 3.3. In Vitro Studies

From the in-silico studies, compounds **11** and **20** were selected to study their ability to be used as novel anticancer agents via the inhibition of mitochondrial complex I-associated hexokinase. 

#### 3.3.1. Cytotoxic Effect of Compounds **11**, **20**, and Tamoxifen (TAM)

In research on new anticancer agents, the most common screening methods are testing against a panel of different cancer cell lines. In this study, an MTT assay was carried out to determine the cytotoxic effect of compounds **11** and **20** on the proliferation of MCF-7, MDA-231, PCL, and the toxicity limit on the normal cell line WISH after 48 h (Figure 3). Compounds **11** and **20** exhibited significant cytotoxic effects on MCF-7, MDA-231, and PCL cancer cell lines as the IC_50_ values of compound **11** were 64.05 ± 0.14 μg/mL (0.135 mM), 77.35 ± 0.09 μg/mL (0.163 mM), and 73.97 ± 0.15 μg/mL (0.156 mM), respectively. The IC_50_ values of compound **20** against MCF-7, MDA-231, and PCL cell lines were 54.41 ± 0.08 μg/mL (0.195 mM), 46.29 ± 0.09 μg/mL (0.166 mM), and 60.79 ± 0.1 μg/mL (0.218 mM), respectively. According to our results, compounds **11** and **20** had more significant inhibitory effects on MCF-7 and MDA-231 breast cancer cell lines respectively as compared with TAM as a reference drug (IC_50_ = 48.35 ± 0.07 μg/mL (0.130 mM) and 44.69 ± 1.1 μg/mL (0.120 mM), respectively). Moreover, compounds **11** and **20** showed lower cytotoxic effects on WISH normal cells (IC_50_ = 116 ± 1.6 μg/mL (0.24 mM) and 113.6 ± 1.7 μg/mL (0.408 mM), respectively). This means that they are much more effective on cancer proliferative cells without any toxic effects on normal cells. This is in accordance with the results of in silico docking studies for these compounds. In contrast, TAM has high cytotoxicity on normal cells (IC_50_ = 30.62 ± 0.21 μg/mL (0.082 mM). Thus, MCF-7 and MDA-231 were selected for further analysis.

#### 3.3.2. Detection of Cell Cycle Arrest Phase

Cell cycle arrest phases are the end step that results from the inhibition of mitochondrial complex I and the activation of AMPK protein kinases. Thus, to determine whether the inhibitory effects of compounds **11** and **20** on MCF-7 and MDA cell lines were associated with cell cycle arrest phases, we performed flow cytometry in a dose-dependent manner for each compound (1/2 IC_50_, IC_50_, 2IC_50_). In the MCF-7 cell line (compound **11**) and MDA cell line (compound **20**), an increase in the percentage of cells in the sub-G_0_/G_1_ phase was observed in cells at all treated doses compared with the findings in untreated cells. (This is the phase at which cells wait to enter the cell cycle to replicate; when the number of cells in this phase increases, this means that the cell cycle has been arrested and division and replication cannot occur.) The 1/2 IC_50_, IC_50_, and 2 IC_50_ of compound **11** and compound **20** showed cell cycle arrest at rates of 12.5%, 17.6%, 25.7% and 16.8%, 31.3%, 41.8%, respectively, at sub-G_0_/G_1_ phase compared with untreated MCF-7 cells (5.6%) and MDA-231 cells (10.8%). Therefore, the overall rates of cell cycle arrest in groups treated with compounds **11** and **20** were 87.6%, 82.5%, 74.7% and 83.2%, 68.7%, 58.5%, respectively, compared with untreated MCF-7 cells (94.4%) and MDA-231 cells (89.2%), as illustrated in Figure 4 and Figure 5. Collectively, these results show the improvement in the ability of compounds **11** and **20** to induce the inhibition of mitochondrial complex I, activate adenosine monophosphate-activated protein kinase (AMPK) and cause apoptotic cell death by arresting the cell cycle in sub-G_0_/G_1_ phase in a dose-dependent manner [3].

#### 3.3.3. Altered mRNA Expression of Apoptosis Markers

The mRNA expression of both MCF-7 and MDA-231 cell lines was quantified by qRT-PCR for Bax, p53 (apoptotic markers), and Bcl-_2_ (anti-apoptotic marker) genes. Bax and p53 were significantly (*p* < 0.0001) upregulated in cells treated with compounds **11** and **20** in a dose-dependent manner, with maximum expression in the double IC_50_ of cells treated with each compound compared with the findings in untreated cells. The Bcl-_2_ gene was significantly (*p* < 0.0001) downregulated in the cells treated with compounds **11** and **20** in a dose-dependent manner, with minimal expression in the double IC_50_ of cells treated with each compound compared with untreated cells, as shown in Figure 6. Thus, the upregulation of Bax and downregulation of Bcl-_2_ in our results mean that compounds **11** and **20** cause mitochondrial membrane dysfunction, while the upregulation of p53 results from the activation of AMPK. This clarifies the ability of compounds **11** and **20** to inhibit mitochondrial complex I, which is the step that leads to cell cycle arrest and finally induces apoptosis [3,44].

#### 3.3.4. Inhibition of Warburg Effect through Hexokinase (ELISA)

It is known that cells depend on the continuous intake of glucose for proliferation and survival. This is especially the case for cancer cells, which prefer aerobic glycolysis for ATP production if they are under hypoxic conditions (Warburg effect). The Warburg effect has thus become a novel target for anticancer therapy. Adenosine monophosphate activated protein kinase (AMPK) is a cellular energy receptor that is phosphorylated in response to energy stress (glucose deprivation and decreased cellular ATP/ADP ratios), leading to the inhibition of mTOR that in turn results in the Warburg effect glycolytic pathway to alter this stress and produce ATP. The Warburg effect can be diminished by inhibiting the hexokinase enzyme, the key enzyme for starting glucose consumption in the glycolytic pathway. Our results significantly showed the inhibition of hexokinase activity in MCF-7 (by compound **11**) and MDA-231 (by compound **20**) cell lines after treatment for 48 h in a dose-dependent manner, as shown in Figure 7. This investigation was performed to confirm their mechanical pathway in stopping cancer proliferation under a decreased level of ATP, which results from the inhibition of mitochondrial complex I [3,45,46].

#### 3.3.5. Oxidative and Antioxidant Biomarkers

Our results showed a significant increase in the MDA level in MCF-7 cells treated with compound **11** and MDA-231 cells treated with compound **20**. On the other hand, our results indicated a significant decrease in the GSH level in MCF-7 cells treated with compound **11** and MDA-231 cells treated with compound **20** in a dose-dependent manner, as shown in Figure 8. This means that compounds **11** and **20** could induce apoptosis in cancer cells by triggering intracellular ROS and inhibiting endogenous antioxidant enzymes. Increased production of ROS in cancer cells leads to the activation of AMPK protein kinase and also causes mitochondrial membrane dysfunction, which finally induces apoptosis and arrests cell survival and proliferation [47,48,49].

#### 3.3.6. Immunoblotting Confirms the AMPK/mTOR Pathway

Immunoblotting results showed that compounds **11** and **20** cause a significant increase in AMPK protein kinase folding with a remarkable decrease in mammalian target of rapamycin (mTOR) folding in MCF-7 and MDA-231 breast cancer cell lines in a dose-dependent manner compared with untreated cells, as shown in Figure 9 and Figure 10. These results confirm the mechanical pathways of compounds **11** and **20** to induce apoptosis. The first pathway is that compounds **11** and **20** cause inhibition of mitochondrial complex I, which in turn causes a decrease in ATP production and an increase in the intracellular ROS generation, leading to activation of AMPK through phosphorylation. Once AMPK becomes phosphorylated, it activates P53, leading to cell cycle arrest. It also inactivates mTOR, leading to the Warburg effect, which is also blocked by compounds **11** and **20**. The second pathway involves the ability of compounds **11** and **20** to act on ROS generation, leading to mitochondrial membrane dysfunction and in turn causing an increase in Bax and a decrease in Bcl-_2_, thereby increasing cytochrome C and caspase 3. These mechanical pathways induce apoptosis and arrest cell survival and proliferation [3,44,45,46,47,48,49].

## 4. Conclusions

Collectively, as shown in Figure 11**,** in this study the synthesized compound **11** (quinazoline amino acid Schiff base) and compound **20** (indole amino acid Schiff base) were selected according to their in silico molecular binding energy toward NADH oxidoreductase mitochondrial complex I-associated hexokinase, as shown in docking studies. Subsequently, in vitro studies confirmed that the selected compounds **11** and **20** can cause apoptosis and cell death through the induction of ROS generation-mediated inhibition of mitochondrial complex I-associated hexokinase. The cellular mechanism by which the interdependence between AMPK activation and mTOR inhibition, p53 activation, and cell cycle arrest blocks the Warburg effect involved in MCF-7 and MDA-231 breast cancer cell lines has been elucidated. Thus, the biological results, as well as the molecular docking studies on compounds **11** and **20**, led us to consider these molecules as promising compounds for future investigation and development of anticancer agents.

## Data Availability

Data is contained within the article and the Appendix A.

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
