# Peer review of "New Amino Acid Schiff Bases as Anticancer Agents via Potential Mitochondrial Complex I-Associated Hexokinase Inhibition and Targeting AMP-Protein Kinases/mTOR Signaling Pathway"

_molecules, 2021, doi:10.3390/molecules26175332_

Round 1

Reviewer 1 Report

In this manuscript titled “New amino acid Schiff bases as anticancer agents via potential mitochondrial complex I-associated hexokinase inhibition and targeting AMP-protein kinases /mTOR signaling pathway”, Noser et al. synthesized and evaluated the novel quinazolinone and indole amino acid Schiff bases for their potency as anticancer agents. Based on their in silico molecular docking studies and in vitro studies, authors claim that these novel compounds can cause apoptosis and cell death through the induction of ROS generation-mediated inhibition of mitochondrial complex I-associated hexokinase. The in vitro studies using MCF-7 and MDA-231 breast cancer cell lines specifically elucidated the cellular mechanism by which the interdependence between AMPK activation and mTOR inhibition, p53 activation, and cell cycle arrest blocks the Warburg effect.

Cancer drug resistance and a lack of sensitivity of tumor cells to such drugs is a widespread concern. The authors developed novel amino acid Schiff bases by utilizing in silico molecular docking studies and investigated their mechanism of action using invitro studies for safe and effective treatment of cancer. The experiments done were mostly adequate for addressing the questions. However, the following minor issues shall be addressed before the acceptance of this manuscript:

  1. In figure 3, compounds 11 and 20 seem to have lower cytotoxic effects on WISH normal cells. Is there any statistic support?
  2. The in vitro studies were mostly done with breast cancer cell lines, although the compounds were shown to have cytotoxicity effects on pancreatic cancer cell line PCL. The authors may discuss the cancer-type specificity of the compounds.
  3. Are in vivo studies planned to test these novel compounds? 
  4. Please add company name with catalog numbers to make it easier for reference in materials & methods section.
  5. In figure 4, values on few figures are not visible clearly (E02 control MCF7, E03 half IC50 PI, E01 2 IC50). Can improve them.

Author Response

Reviewer: 1

               We would like to thank reviewer # 1 for his effort in providing these comments.

  • In figure 3, compounds 11 and 20 seem to have lower cytotoxic effects on WISH normal cells. Is there any statistic support?

Yes, the experimental cytotoxic effects data on all cell lines either cancer or normal cells were done in triplicate and expressed as mean ± SE, but the SE was too small to be seen as an error bar in Figure 3. Also, the (IC50) was calculated using the sigmoidal curve with GraphPad (Prism) statistical software.

  • The in vitro studies were mostly done with breast cancer cell lines, although the compounds were shown to have cytotoxicity effects on pancreatic cancer cell line PCL. The authors may discuss the cancer-type specificity of the compounds

            In research on new anticancer agents, the most common screening methods were tested against a panel of different cancer cell lines, and we have chosen only one of these cell lines that gives the lowest IC50 for the two compounds to continue on it the other further biochemical studies to elucidate the potential anticancer cellular mechanism on these novel compounds, as this research article was not funded at all. So, it was too hard and more expensive to do all biological applications on these three cancer cell lines. Regards this reason and according to our results, compounds 11 and 20 showed more significant inhibitory effects on MCF-7 and MDA-231 breast cancer cell lines respectively than PCL pancreatic cancer cell line as showed in the Table below.

             To be clear:

Table: Calculating IC50 in the cancer cell lines

IC50 after 48h

MCF-7

MDA-231

PCL

Compound 11

64.05 mg/ml (0.135 mM)

77.35 mg/ml (0.163 mM)

73.97 mg/ml (0.156 mM)

Compound 20

54.41 mg/ml (0.195 mM)

46.29 mg/ml (0.166 mM)

60.79 mg/ml (0.218 mM)

  • Are in vivo studies planned to test these novel compounds?

Yes, we planned to study the anticancer potency of these two novel compounds on xenograft mice (in vivo). We will study that in a future work. 

  • Please add company name with catalog numbers to make it easier for reference in materials & methods section.

    The company name with catalog numbers was already added in the materials & methods section.

  • In figure 4, values on few figures are not visible clearly (E02 control MCF7, E03 half IC50 PI, E01 2 IC50). Can improve them

In Figure 4, values were improved according to the reviewer's comment.

Reviewer 2 Report

In “New amino acid Schiff bases as anticancer agents via potential mitochondrial complex I-associated hexokinase inhibition and targeting AMP-protein kinases /mTOR signaling pathway” Ahmed A. Noser et al. synthesized two series of novel amino acid Schiff base ligands containing heterocyclic moieties such as quinazolinone 3–11 and indole 12–20. From the in silico studies, compounds 11 and 20 were selected to study their ability to be used as novel anticancer agents via the inhibition of mitochondrial complex I-associated hexokinase. compounds 11 and 20 had more significant inhibitory effects on MCF-7 and MDA-231 breast cancer cell lines respectively as compared with TAM as a reference. Moreover, compounds 11 and 20 showed lower cytotoxic effects on WISH normal cells. So they are much more effective on cancer proliferative cells without any toxic effects on normal cells. Compounds 11 and 20  induce the inhibition of mitochondrial complex I, activate adenosine monophosphate activated protein kinase (AMPK), and cause apoptotic cell death by arresting the cell cycle in sub-G0/G1 phase in a dose-dependent manner. Moreover, compounds 11 and 20 showed significant inhibition of mTOR protein, which led to the activation of aerobic glycolysis for survival. This alternative pathway was also blocked as compounds 11 and 20 showed significant inhibitory effects on the hexokinase enzyme. Compounds 11 and 20 can be regarded as promising compounds for the development of anticancer agents.

The manuscript is interesting and well written, however the authors

  • should substitute the term SHEET with LAYER in the paragraph 2.3.6.;
  • should indicate the concentrations of the primary antibodies used in the paragraph 2.3.7.;
  • should use capital letters to indicate each graph in figures 6, 8, 9 and 10.
  • Why was the MTT test performed to determine the cytotoxic effect of compounds 11 and 20 on the proliferation of MCF-7, MDA-231, PCL and toxicity limit on the normal WISH cell line only after 48 h? (Paragraph 3.3.1).

Author Response

Reviewer: 2

Comments:

                 We would like to thank reviewer # 2 for his effort in providing these comments.

  • The authors should substitute the term SHEET with LAYER in the paragraph 2.3.6.

The term SHEET was substituted with LAYER in paragraph 2.3.6.

  • should indicate the concentrations of the primary antibodies used in the paragraph 2.3.7.

The concentrations of the primary antibodies used were added in paragraph 2.3.7.

  • should use capital letters to indicate each graph in figures 6, 8, 9 and 10.

            Capital letters were added to Figures 6, 8, 9 and 10.

  • Why was the MTT test performed to determine the cytotoxic effect of compounds 11 and 20 on the proliferation of MCF-7, MDA-231, PCL and toxicity limit on the normal WISH cell line only after 48 h? (Paragraph 3.3.1).

The optimal incubation time in anticancer activity study depends on the drug ‘s mechanism of action and the cancer cell line used. According to this view, 48h incubation was sufficient to see both survival and anti-proliferation effect. Moreover, we have performed the MTT test after 48h according to the literature of other authors that used near to the structure of our novel compounds on different cancer cell lines as references below.

For quinazoline compounds as example reference papers:

  1. De, A.K., Muthiyan, R., Mondal, S., Mahanta, N., Bhattacharya, D., Ponraj, P., Muniswamy, K., Kundu, A., Kundu, M.S., Sunder, J. and Karunakaran, D., 2019. A natural quinazoline derivative from marine sponge hyrtios erectus induces apoptosis of breast cancer cells via ROS production and intrinsic or extrinsic apoptosis pathways. Marine drugs, 17(12), p.658.

  1. Noser, A.A., El-Naggar, M., Donia, T. and Abdelmonsef, A.H., 2020. Synthesis, in silico and in vitro assessment of new quinazolinones as anticancer agents via potential AKT inhibition. Molecules, 25(20), p.4780.

  1. Hassanzadeh, F., Sadeghi-Aliabadi, H., Jafari, E., Sharifzadeh, A. and Dana, N., 2019. Synthesis and cytotoxic evaluation of some quinazolinone-5-(4-chlorophenyl) 1, 3, 4-oxadiazole conjugates. Research in pharmaceutical sciences, 14(5), p.408.

For Indole compounds as example reference papers:

  1. Lee, J.Y., Lim, H.M., Lee, C.M., Park, S.H. and Nam, M.J., 2021. Indole-3-carbinol inhibits the proliferation of colorectal carcinoma LoVo cells through activation of the apoptotic signaling pathway. Human & Experimental Toxicology, p.09603271211021475.

  1. Sever, B., Altıntop, M.D., Özdemir, A., Akalın Çiftçi, G., Ellakwa, D.E., Tateishi, H., Radwan, M.O., Ibrahim, M.A., Otsuka, M., Fujita, M. and Ciftci, H.I., 2020. In Vitro and In Silico Evaluation of Anticancer Activity of New Indole-Based 1, 3, 4-Oxadiazoles as EGFR and COX-2 Inhibitors. Molecules, 25(21), p.5190.

         Finally: We thank the reviewers for their help in increasing the quality of this paper.